# Diagnosis and Therapy of Soft Tissue Sarcomas: Spanish Group for Research in Sarcomas (GEIS) Guidelines

**DOI:** 10.3390/cancers17193158

**Published:** 2025-09-29

**Authors:** Maria Angeles Vaz-Salgado, Claudia Valverde-Morales, Rosa Alvarez, Jose Manuel Asencio, Erica Collado, Enrique de Alava, Roberto Diaz Beveridge, M. Carmen Gómez-Mateo, Isidro Gracia Alegria, Gloria Marquina, Javier Martin Broto, Javier Martínez-Trufero, José Antonio Narváez, Andres Redondo, Ana Sebio, Ramona Verges, Joan Maria Viñals, Xavier García del Muro

**Affiliations:** 1Medical Oncology Department, Hospital Universitario Ramón y Cajal-Irycis, 28034 Madrid, Spain; 2Medical Oncology Department, Hospital Universitario Vall d’Hebron, 08035 Barcelona, Spain; cvalverde@vhio.net.medical; 3Medical Oncology Department, Hospital General Universitario Gregorio Marañón, 28007 Madrid, Spain; ralvareza@salud.madrid.org; 4General Surgery Department, CSUR Sarcoma and Mesenchymal Tumors, Hospital General Universitario Gregorio Marañón, 28007 Madrid, Spain; josemanuel.asensio@salud.madrid.org; 5Radiation Oncology Department, Hospital Clínico Valencia, 46010 Valencia, Spain; collado_eri@gva.es; 6Institute of Biomedicine of Sevilla (IBiS), Virgen del Rocio University Hospital/CSIC/University of Sevilla/CIBERONC, 41013 Seville, Spain; enrique.alava.sspa@juntadeandalucia.es; 7Department of Normal and Pathological Cytology and Histology, School of Medicine, University of Seville, 41004 Seville, Spain; 8Medical Oncology Department, Hospital Universitario y Politécnico La Fe de Valencia, 46026 Valencia, Spain; diaz_rob@gva.es; 9Pathology Department, Hospital Universitario Miguel Servet, 50009 Zaragoza, Spain; mcgomezm@salud.aragon.es; 10Orthopaedic Oncology Unit, Orthoapedic and Traumatology Department, Hospital Sant Pau, 08041 Barcelona, Spain; igracia@santpau.cat; 11Medical Oncology Department, Hospital Clinico San Carlos, School of Medicine, UCM, IdISSC, EURACAN Referral Centre, 28040 Madrid, Spain; gloria.marquina@salud.madrid.org; 12Medical Oncology Department, Hospital Universitario Fundación Jiménez Díaz, 28040 Madrid, Spain; jmartin@atbsarc.org; 13General de Villalba, 28400 Madrid, Spain; 14Instituto de Investigación Sanitaria Fundación Jiménez Díaz, 28003 Madrid, Spain; 15Medical Oncology Department, Hospital Universitario Miguel Servet, 50009 Zaragoza, Spain; jmtrufero@seom.org; 16Musculoskeletal Radiology, Hospital Universitario de Bellvitge, 08907 Barcelona, Spain; jnarvaez@bellvitgehospital.cat; 17Medical Oncology Department, Hospital Universitario La Paz-IdiPAZ, 28046 Madrid, Spain; andres.redondos@uam.es; 18Medical Oncology Department, Hospital de la Santa Creu i Sant Pau, 08041 Barcelona, Spain; asebio@santpau.cat; 19Radiotherapy Department, Hospital Universitario Vall d’Hebron, Universitat Autònoma de Barcelona, 08193 Barcelona, Spain; rverges@vhebron.net; 20Plastic Surgery Department, Hospital Bellvitge, Hospitalet de Llobregat, 08908 Barcelona, Spain; jm.vinals@bellvitgehospital.cat; 21Sarcoma Multidisciplinary Unit, Medical Oncology Department, Institute Catalan of Oncology, University of Barcelona, 08007 Barcelona, Spain; garciadelmuro@iconcologia.net

**Keywords:** soft tissue sarcoma, clinical guidelines, sarcoma multidisciplinary therapy

## Abstract

Soft tissue sarcomas are rare, heterogeneous tumors requiring management in expert, multidisciplinary reference centres. Early suspicion, appropriate imaging (MRI as first choice), and core needle biopsy performed in referral centres are critical for diagnosis. Surgery with clear margins is the cornerstone of localized treatment, complemented by radiotherapy and chemotherapy in selected cases. Treatment of advanced or metastatic disease should be tailored to histologic subtype, with systemic therapy options including anthracyclines, ifosfamide, trabectedin, gemcitabine combinations, eribulin and targeted agents. Molecular diagnostics and participation in clinical trials are strongly encouraged to improve outcomes.

## 1. Introduction

Soft tissue sarcomas (STSs) are a rare and heterogeneous group of tumours of mesenchymal cell origin, with an estimated incidence of approximately five cases per 100,000 per year in Europe. STSs comprise more than 50 histologic subtypes, as recognized by the World Health Organization (WHO). Among the most relevant subtypes are liposarcomas, leiomyosarcomas, synovial sarcomas, malignant peripheral nerve sheath tumours, angiosarcomas, rhabdomyosarcomas, and undifferentiated pleomorphic sarcomas. These entities often differ substantially in their biology, clinical course, prognosis, and response to therapy. This framework highlights both the rarity and the complexity of this group of diseases, which pose significant challenges in diagnosis and management. Effective care requires specialized expertise and a multidisciplinary approach, ideally within high-volume reference centres.

## 2. Methodology

These guidelines were developed by a multidisciplinary group of expert specialists in the various fields involved in sarcoma diagnosis and therapy. Firstly, the three coordinating authors (MV, CV, and XG) formulated a set of key questions covering the main aspects of STS diagnosis and treatment. These questions were then assigned to experts in the respective fields for further development. A comprehensive literature search was conducted in the PubMed database and the Cochrane Library to identify published studies providing evidence related to the proposed questions. Additionally, relevant yet unpublished clinical studies on STS therapy, presented at the American Society of Clinical Oncology (ASCO) and European Society for Medical Oncology (ESMO) meetings, were reviewed. The search was restricted to human clinical trials, meta-analyses, and consensus statements, and, in areas lacking prospective studies, observational studies were included.

During a consensus meeting, the experts assigned to each question presented their findings to the entire group for discussion. The group critically reviewed the evidence and formulated guideline recommendations, assigning a level of evidence and a grade to each recommendation according to the Infectious Diseases Society of America guidelines [1].

The three coordinating authors contributed to the preparation of the draft guideline document. All authors reviewed and approved the final version. Panel members completed a disclosure form, outlining any financial or other relevant relationships pertinent to the guideline.

## 3. Recommendations

### 3.1. Section 1: Diagnostic Approach

#### 3.1.1. Question: What Are the Key Warning Signs to Rule out Sarcomas, and How Should We Proceed?

Recommendations:There are four warning signs for suspicion of sarcomas (IV,A):
-Mass > golf ball (4.3 cm).-Recent increase in tumour size.-Depth location.-Pain.Sarcoma early recognition and referral to a reference/expert centre, preferably before biopsy, are crucial since the first approach is critical for functional results (III,A).

Literature review and interpretation:

Symptoms of presentation of benign and malignant soft tissue tumours can frequently overlap. A growing soft tissue mass with a diameter over 4.3 cm, irrespective of pain association, is a suspected sarcoma. The location could be deep or superficial. The more these clinical features are present, the greater the risk of malignancy. The best individual indicator of malignancy is increased size [2,3]. However, one-third of sarcomas are found accidentally [4].

As soon as a sarcoma is suspected, the patient should be referred to a sarcoma reference or expert centre, even before biopsy. The multidisciplinary diagnosis and therapeutic approach and the annual volume of sarcoma cases in a sarcoma referral centre seem to be critical for local control, morbidity, and survival of patients [5,6]. Centralization of patients with sarcomas has long been recognized as an important factor in improving clinical outcomes. The definition of a sarcoma referral centre is based on an expert multidisciplinary specialist team that holds regular committee sessions with, at least, pathologists, radiologists, surgeons, and radiation and medical oncologists, with wide experience in sarcoma management [7].

#### 3.1.2. Question: What Imaging Studies Should Be Performed When an STS Is Suspected?

Recommendations:Magnetic resonance imaging (MRI) is the modality of choice for the diagnosis and local staging of soft tissue sarcomas of extremities, trunk wall, pelvis, and head/neck (IV,A). MRI should be performed with intravenous contrast administration, and it is mandatory to obtain images in at least two planes (IV,A).When MRI is contraindicated, computed tomography (CT) with intravenous contrast should be performed, preferably with sagittal and coronal reconstructions (IV,A). CT can also be appropriate as a modality of choice in retroperitoneal/intra-abdominal and pleuropulmonary sarcomas (IV,A).Ultrasound (US) may be used as first-line imaging in cases of superficial, small lesions, but if the lesion is considered indeterminate, additional MRI or CT should be performed (III,A).

Literature review and interpretation:

The addition of intravenous contrast to non-enhanced MRI improves differentiation between benign and malignant soft tissue lesions [8]. MRI should be performed before biopsy to avoid post-biopsy oedema or haemorrhage that may affect the characteristics and extent of the tumour.

Hung et al. prospectively evaluated the accuracy of US in the assessment of superficial soft tissue masses. US distinguishes benign from malignant superficial soft tissue masses with a sensitivity of 93.3%, specificity of 97.9%, positive predictive value of 45.2%, and negative predictive value of 99.9% [9].

#### 3.1.3. Question: What Essential Data Should Be Included in the MRI Report?

Recommendations:The MRI report should describe the following lesion characteristics (IV,B): -Size (measured in all three dimensions).-Location in relation to the fascia (superficial or deep). Revisar inglés-Compartment/s involved (muscle involvement and extension through the fascia and to the skin).-Relationship with neurovascular structures.-Bone involvement (cortical and marrow space).-Extension of perilesional oedema/perilesional enhancement.-The pattern of contrast enhancement (identification of necrosis).-Suggestions of areas for biopsy.MRI signs that suggest the diagnosis of a sarcoma include a size larger than 5 cm, a deep location relative to the fascia, and heterogeneous signal intensity/contrast enhancement. Since there is some overlap in the MRI characteristics between STSs and some benign soft tissue tumours, it is important to note that any lesion that cannot be unequivocally characterized by MRI as benign should be considered “indeterminate” and requires biopsy. The approach to such indeterminate lesions is that they are sarcomas until proven otherwise (IV,B).

Literature review and interpretation:

In a large prospective study, MRI reads by expert radiologists in consensus identified malignant lesions with a sensibility of 93%, specificity of 82%, negative predictive value of 98%, and positive predictive value of 60%; the exact histology of the tumour was predicted in 50% of cases [10]. According to Holzapfel et al., encasement of neurovascular structures should be diagnosed if the contact between a soft tissue sarcoma and vascular or neural circumference exceeds 180° [11]. The extent of perilesional oedema detected on MRI is important for treatment planning because the presence of viable tumour cells has been demonstrated in this area, beyond the margin of the lesion [12]. In the same way, peritumoral enhancement, defined as contrast enhancement beyond the apparent tumour borders, is associated with grade III in histology [13,14]. A recent international collaborative group study demonstrates that identification of necrosis by imaging methods may improve the distinction between low- and high-grade soft tissue sarcoma since radiologists tend to avoid necrotic areas when performing core needle biopsies [15].

#### 3.1.4. Question: Which Modality of Biopsy Should Be Performed?

Recommendations:Core needle biopsy (CNB) should be the standard procedure for any suspicious sarcoma lesion (IV,A): -Needles of 14–16 G should be used.-Several cores must be taken. A minimum of 3–4 cores for diagnosis purposes.A total of 4–6 cores for research purposes/biobanking.Planification of the site of the biopsy is mandatory (IV,A): -It should preferably be performed in the same hospital where the surgery is going to be performed.-It should be guided by images.-The biopsy tract should be included in the final surgical specimen.Excisional biopsy is possible when dealing with small (<3 cm) superficial lesions. Incisional biopsies (IBs) must be relegated to exceptional cases. Fine needle aspiration (FNA) is not generally recommended (IV,A).

Literature review and interpretation:

CNB should be considered the first technique to diagnose STSs [16]. This technique represents a great improvement in terms of minimal morbidity, less cost and time consumption, and limited tumour spread [17,18]. Although controversial results have been reported, a recent meta-analysis demonstrated the high accuracy of CNB with fewer complications compared with IBs [19]. In retroperitoneal sarcomas, CNB is also recommended for preoperative assessment [20].

Most international guidelines and protocols from referral centres suggest at least 3 passes for CNB with ≥14–16-gauge needles [16,19]. No differences between 14–16 G versus 18–22 G needles were demonstrated in retroperitoneal sarcomas [20] but no prospective trials or meta-analysis results regarding the number of passes and the size of needles have been published [19]. CNB could be carried out by a surgeon or a radiologist under radiological guidance to improve efficiency and safety [21,22,23,24,25,26]. Although the risk of needle tract seeding (NTS) seems to be <1% [27], the biopsy tract should be included in the surgical resection specimen except for retroperitoneal sarcomas. Because of this, it is strongly recommended to plan the biopsy site [16,19,21,27].

Limited samples are quite challenging even for experienced pathologists [28]. When the result of CNB is non-diagnostic, an open biopsy could be an option [21,22,23]. FNA is not generally recommended except for in selected clinical situations where only a confirmation of recurrence/metastases is needed [16,21,23,29].

#### 3.1.5. Question: What Essential Information Should the Pathological Report Include for Diagnostic Biopsies and Surgical Resection Specimens?

Recommendations:Core biopsy/incisional biopsy (IV,A): -Biopsy procedure.-Site and depth (based on clinical information).-Histological subtype (according to the current WHO classification). When non-diagnostic: descriptive category (pleomorphic, spindle, epithelioid, or round cell sarcomas).-Histological grade (according to the FNCLCC). Sometimes only a provisional grade can be provided in core biopsies (“at least grade”).Some sarcoma subtypes deserve a specific grade with the histological diagnosis (grade histological subtype-dependent).Some others do not fit into the FNCLCC system.-Mitotic rate.-Percentage of necrosis.-Immunohistochemistry and molecular tests performed.Surgical specimen (IV,A): -Surgical procedure and orientation (indicated by the surgeon).-Neoadjuvant treatment.-Macroscopic description.-Site and depth (tissue plane involvement).-Size (largest diameter).-Histological subtype (according to the current WHO classification).-Histological grade (according to the FNCLCC and specifying each of the 3 items). Sarcomas are not gradable after neoadjuvant treatment.-Mitotic rate.-Percentage of necrosis.-Neoadjuvant treatment effect.-Immunohistochemistry and molecular tests performed.-Involvement of any structure (fascia, nerves, vessels, muscle, organs…).-Lymph node involvement (only in a few subtypes).-Margins (a microscopic distance to all margins that is closer to 20 mm) with the description of any anatomic barrier presented between the tumour and margins (fascia, periosteum…).

Literature review and interpretation:

Pathology plays a crucial role in treating sarcoma patients, especially with the increasing use of neoadjuvant therapies in adult soft tissue sarcomas (STSs). It guides diagnosis, patient selection for targeted treatments, and evaluation of tumour response. To improve the consistency and quality of pathological reports, diverse scientific societies have focused on and implemented their “pathological protocols” [30,31,32,33]. Herein, we have reviewed and updated them:-Diagnostic biopsy: The WHO classification for STSs should be applied. If not, a descriptive category (pleomorphic, spindle, epithelioid, or round cell sarcomas) should be used. The mitotic count should be assessed in the most active area, in 10 successive HPFs or a mitotic count per mm [34,35].Histological grade is a key prognostic factor, with the FNCLCC system recommended for grading. A provisional grade may be reported if necessary. Modifications using Ki67 or radiological evaluation could be considered [36,37,38].Exceptions to the FNCLCC system [39]: ○Atypical lipomatous tumours/well-differentiated liposarcomas, dermatofibrosarcoma protuberans, and infantile fibrosarcomas are considered Grade 1 by definition.○Ewing sarcomas and other undifferentiated round cell sarcomas, rhabdomyosarcomas (all types), angiosarcomas, pleomorphic liposarcomas, soft tissue osteosarcomas, mesenchymal chondrosarcomas, desmoplastic small round cell tumours, extra-renal rhabdoid tumours, and intimal sarcomas are defined as Grade 3.○Other examples such as alveolar soft part sarcomas, clear cell sarcomas, epithelioid sarcomas, low-grade fibromyxoid sarcomas, and sclerosing epithelioid sarcomas are not graded. Synovial sarcomas can be included in this group, although some studies have shown differences between G2 and G3 using the FNCLCC system.○Histological subtypes which do not fit into the FNCLCC system deserve specific risk models or high-grade criteria: myxoid liposarcomas and solitary fibrous tumours (SFTs).○Grading malignant peripheral nerve sheath tumours may be prognostic, but this is currently being debated.○Grading parameters are not yet well defined to predict behaviour for extraeskeletal myxoid chondrocarcoma and epithelioid haemangioendothelioma (EHE).

It is encouraged not to grade intermediate behaviour sarcomas, meaning locally aggressive or rarely metastasizing ones.

-Surgical specimen. Histological subtype remains the main factor to predict outcome and sensitivity to therapy in sarcomas [40,41]. Grade, size, vascular invasion, necrosis, and growth pattern have been identified as prognostic factors [42,43]. The clearance of the excision margin is an important factor in predicting local recurrences [44,45]. After neoadjuvant therapy, histological grading is not applicable. Therapy-related changes, like necrosis and fibrosis, can complicate the assessment. The EORTC-STBSG response score suggests evaluating stainable tumour cells, though other changes, like fibrosis, may also have prognostic value [46,47].Any involvement of surrounding structures (e.g., fascia and nerves) should be reported. Lymph node involvement is rare [48]. The most common subtypes which can spread to the lymph nodes are synovial sarcomas, epithelioid sarcomas, clear cell sarcomas, rhabdomyosarcomas, and angiosarcomas.The surgical procedure and margin status should be clearly indicated. Microscopic distance to margins should be measured, especially if close to 20 mm, with a specific description of any anatomic barrier presented at those margins (fascia, periosteum…) [44,45].Histological subtype remains crucial for predicting outcomes, along with other factors, like size, vascular invasion, and necrosis [40,41]. Excision margin clearance is important for predicting local recurrences [44,45].

#### 3.1.6. Question: When Should Molecular Studies and NGS/High-Throughput Molecular Technologies Be Performed?

Recommendations:Morphology and immunohistochemistry, along with clinical information, are usually enough to guarantee an accurate diagnosis when they show typical characteristics. A wide range of surrogate markers of molecular alterations is available (IV,A).Molecular analyses are useful (IV,A): -When the neoplasm presents atypical characteristics (uncommon clinical setting, overlapping morphology, or doubtful immunohistochemistry).-In small round cell tumours.-To label entities currently defined by a distinctive molecular aberration.-For prognostic stratification.-To find predictive biomarkers.-For clinical trial enrolment.Indications for NGS are currently being described and may change in the next few years. They include cases in which at least two FISH assays are required to perform diagnosis/differential diagnosis of a given case (IV,A).

Literature review and interpretation:

Contemporary sarcoma evaluation should integrate morphological assessment with immunohistochemical testing and relevant cytogenetic and molecular techniques [49]. Many sarcomas exhibit characteristic molecular abnormalities that are already used as diagnostic tools. The latest WHO classification of soft tissue and bone tumours includes key molecular findings, refining the classification by integrating morphology and genetic aberrations [34,50]. Some molecular abnormalities have led to the identification of new entities defined by recurrent gene fusions, such as NTRK-rearranged tumours or SMAD3-positive fibroblastic tumours [50].

Despite the availability of new markers, some challenging cases require molecular studies for crucial diagnostic information, especially in undifferentiated small round cell sarcomas, like Ewing’s sarcoma and its variants [51]. Moreover, some genetic alterations, such as PAX-FKHR rearrangement in rhabdomyosarcomas, can have a significant prognostic impact [52,53]. Additionally, molecular testing may help to predict therapy response and select patients for targeted therapies, such as those with NTRK-rearranged tumours [54,55,56].

Several molecular technologies are available, including conventional karyotyping, FISH, next-generation sequencing (NGS), RNA-Seq, and RT-PCR [49,50,57]. NGS is particularly useful when more than two FISH analyses are needed. The best contribution of high-throughput genetic technologies is to perform genomically guided treatments which could improve patient care [58], as suggested by retrospective data. However, evidence to support their routine implementation in sarcoma management is still lacking [55,59,60]. New immunohistochemical markers derived from specific genetic alterations offer a robust, cost-effective alternative, making their use widespread in daily practice [34,49].

#### 3.1.7. Question: What Is the Recommended Approach for Staging STSs?

Recommendations:A chest CT scan is indicated in all cases of STS to rule out pulmonary metastases (IV,A).In cases of myxoid liposarcomas, a whole-body MRI study is recommended [IV,B]. If whole-body MRI is not available, thoracoabdominal CT and spine and pelvic MRI are recommended (IV,B).The clinical benefit of performing an abdominal CT scan in cases of epithelioid sarcomas, synovial sarcomas, angiosarcomas, and leiomyosarcomas should be assessed individually (IV,B).Brain MRI should be considered in patients with clear cell sarcomas, angiosarcomas, and alveolar soft tissue part sarcomas (IV,B).FDG PET-CT is recommended for the detection of malignant transformation of neurofibromas in patients with neurofibromatosis type 1 and for distant staging in cases of local recurrence of STSs in which aggressive salvage surgery is considered (III,A).

Literature review and interpretation:

In a retrospective review of 169 cases of myxoid liposarcomas, Visgauss et al. found metastatic disease in 31 patients (20%), mainly located in the soft tissue (84%), lungs (68%), abdomen (48%), solid organs (45%), and bone (45%) [61]. Metastatic disease was identified outside surveillance imaging with thoraco-abdominal CT in 58% of the patients. Of 14 cases with bone metastases, only 1 patient CT identified sclerotic bone lesions. These authors suggest the use of whole-body MRI in the follow-up of patients with myxoid liposarcomas [62].

### 3.2. Section 2: Treatment of Localized Disease

#### 3.2.1. Question: What Is the Appropriate Surgical Procedure for a Patient with a Localized STS?

Recommendations:An en bloc surgery free of tumour margins, causing the least possible functional impairment, is the recommended treatment (II,A).In patients with a prior STS resection with positive margins for the tumour, re-intervention, if feasible, must be considered to expand margins (III,A).However, marginal surgery could be accepted in some selected cases of low-grade STSs, such as atypical lipomatous tumours (III,A).In case of a previous unplanned suboptimal surgery (whoops surgery), re-excision is the first option to be considered.

Literature review and interpretation:

Surgical treatment is usually the main pillar of multimodal therapy for the treatment of STSs. Surgical treatment should be performed by expert surgeons and should always be based on consensus decisions made in a multidisciplinary board. Although STSs are a very heterogeneous group of neoplasms, the main principles of surgery are common to all of them. Some basic surgical principles for limb rescue are the following [63]:Obtaining safe cancer-free margins [64].Resection should take precedence over reconstruction [65].Obtaining adequate soft part coverage [66].

#### 3.2.2. Question: In Which Cases Is Limb Amputation Indicated?

Recommendations:Limb amputation is indicated in the following cases: -When there is a wide infiltration of major neurovascular structures of the limb (II,A).-When free margins, i.e., an R0 or R1 resection planned in a reference centre, cannot be guaranteed (II,A).-When reconstruction and appropriate functional results are not feasible or at least the expected results are similar to those of conservative surgery (II,A).

Literature review and interpretation:

Currently, the past rigid limb amputation criteria have been gradually reviewed, and, thanks to recent advances in surgery, especially in their reconstructive aspects, amputation has been progressively limited to these situations [64,67,68].

#### 3.2.3. Question: What Reconstructive Procedures Are Used in Sarcoma Surgery and in Which Cases?

Recommendations:Reconstruction using free flaps or pedicle flaps provides well blood-supplied tissue, making great resections feasible and facilitating subsequent adjuvant treatments (III,A). They are especially indicated in irradiated areas and head and neck localizations (IV,B).

Literature review and interpretation:

Plastic surgical techniques have significantly advanced the surgical management of STSs, enabling more extensive limb-conserving treatments [69,70]. However, to prevent tumour contamination of donor sites, it is essential to adhere to a strict surgical protocol, as metastases have been reported in donor sites in some cases [71]. It is recommended to incorporate donor site examinations into routine follow-up physical assessments.

#### 3.2.4. Question: In Which Cases Should Perioperative Radiotherapy (RT) Be Administered, and What Is the Preferred Timing (Preoperative or Postoperative) and Dosage?

Recommendations:Pre- and postoperative RT are considered standard approaches for most intermediate or high-grade STSs (usually G2–3, deep tumours, >5 cm) if there is no resection of the entire compartment (II,A).RT is never a substitute for an R0 resection, and re-resection should be considered to achieve negative margins in patients with prior suboptimal surgery, if feasible (IV,A).Pre- and postoperative RT achieve similar outcomes in local control and survival. Although both modalities are acceptable (I,A), preoperative RT would be preferable, especially in large tumours (II,C).Preoperative RT is administered 3 to 6 weeks before surgery with a dose of 50 Gy in 1.8–2 Gy/fraction. Postoperative RT requires higher doses, 50 Gy + 10–16 Gy, in R0 resections and larger fields of treatment. Higher radiation doses should be administered in R1 and R2 resections (II,IA).

Literature review and interpretation:

The addition of RT to wide resections (i.e., free-margin excision) allows for preservation of function with similar local control and survival rates as radical resections (i.e., compartmental excision/amputation). Two randomized trials have demonstrated the local control advantage of adjuvant RT over surgery alone in STSs, one with brachytherapy (BT) and the other with postoperative external beam RT (EBRT) [72,73,74]. In the BT study, an improvement in 5-year local control rates was limited to patients with high-grade lesions (82% vs. 69%) [72]. However, in the EBRT study, a reduction in local recurrence was observed in both high-grade and low-grade extremity tumours [73,74]. A recent meta-analysis comparing EBRT vs. no EBRT in the treatment of resectable STSs [75] indicated that EBRT reduced local relapses of STSs of the extremity, head and neck, or trunk wall (OR 0.49, *p* = 0.001).

Regarding the time of RT, a phase III study showed similar local control and progression-free survival rates (PFSs) with preoperative and postoperative RT in localized primary or recurrent STS patients [76]. While preoperative RT was associated with more acute wound complications (35% vs. 17% for postoperative RT), postoperative RT was associated with a worse long-term functional impact due to fibrosis, oedema, or joint stiffness. Additionally, a recent large observational study showed that both preoperative and postoperative RT were associated with increased overall survival (OS). Interestingly, the study determined that preoperative RT was predictive of R0 surgery [77]. In patients treated with preoperative RT that have a subsequent R1 resection, the administration of a postoperative RT boost does not seem to improve local control [78]. Potential advantages for preoperative RT compared to postoperative RT are a better definition of tumour volume, the need for lower doses and smaller volumes of healthy tissue, and an increased ability to obtain R0 in surgery. Instead, disadvantages include interference with pathology and a higher proportion of acute wound complications.

#### 3.2.5. Question: When Should Neoadjuvant Chemotherapy Be Administered? What Is the Recommended Regimen, and How Can It Be Combined with RT?

Recommendations:Neoadjuvant chemotherapy (NACT), instead of an adjuvant one, is an option of treatment for patients with a high-risk (high grade, deep, and >5 cm or >60% risk of mortality at 10 years estimated by Sarculator) localized STS located in the extremities or trunk wall (I,A). It is specially indicated in large STSs that are marginally resectable or require very aggressive surgery without assuring clean margins (IIIA).Histotype also has to be considered for selecting patients for NACT, being indicated, but not restricted, in the following tumours: high-grade myxoid liposarcomas, leiomyosarcomas, synovial sarcomas, malignant peripheral nerve sheath tumours, and undifferentiated pleomorphic sarcomas (I,A). STS histotypes that are considered chemo-resistant (such as alveolar STSs or clear cell sarcomas) should not be treated with NACT (IV,A).A regimen including anthracycline and ifosfamide is recommended. The highest level of evidence supports 3 cycles of epirrubicin 60 mg/m^2^ on days 1–2 and ifosfamide 3 g/m^2^ on days 1–3 (I,A)The use of an alternative neoadjuvant histotype-driven therapy was not associated with a better DFS or OS (I,A). Only in high-grade myxoid liposarcomas could trabectedin be an alternative to the standard scheme in case of contraindication for anthracycline (II,B).The addition of radiation therapy to NACT could be considered, based upon institutional preference and expertise. The optimal approach has not been established, but concurrent epirrubicin–ifosfamide and RT at a total dose of 50 Gy is feasible (III,B).

Literature review and interpretation:

NACT in STSs could have some potential clinical benefits: improvement of resectability, early treatment of micrometastasis, and reduction in relapses. However, there are no randomized trials that have adequately explored its benefit versus no systemic treatment or adjuvant chemotherapy. A randomized phase II trial that compared three cycles of neoadjuvant chemotherapy with doxorubicin and ifosfamide versus surgery alone in high-risk STSs did not demonstrate any benefit but had the significant limitation of using a suboptimal dose of chemotherapy [79]. Subsequently, a randomized non-inferiority phase III study compared three cycles of preoperative chemotherapy with epirubicin and ifosfamide (60 mg/m^2^ for days 1–2 and 3 g/m^2^ for days 1–3, respectively) versus the same regimen plus two additional adjuvant cycles of treatment after surgery. The results show that three cycles were not inferior to five cycles in terms of recurrence and survival [80]. In a recent phase III study in localized high-grade STSs comparing a standard scheme (epirubicin plus ifosfamide) with histology-tailored chemotherapy for each of the 5 histological subtypes that were included (undifferentiated pleomorphic sarcomas, leiomyosarcomas, synovial sarcomas, myxoid round cell liposarcomas, and malignant peripheral nerve sheath tumours), histotype-tailored NACT was associated with worse OS. Only in high-grade myxoid liposarcomas does trabectedin achieve a DFS and OS similar to the standard regimen [81].

There is a lack of randomized studies evaluating preoperative radiation therapy with NACT. Some old studies explored interdigitated chemo-radiation (CRT), with a conventional dose of radiation given as two split courses in between chemotherapy cycles of the MAID regimen [82]. However, a more recent report describes the feasibility of neoadjuvant concurrent CRT in the setting of a randomized trial with a standard chemotherapy regimen. At the treating physician’s discretion, RT was delivered concurrently with cycles 2 and 3 to a dose of 44 to 50 Gy using 2 Gy fraction sizes. Concurrent CRT was feasible and reduced local relapses in cases with subsequent surgery with an affected margin compared to the exclusive administration of NACT, but this was associated with a higher incidence of grade 4 thrombocytopenia and wound complications [83].

#### 3.2.6. Question: When Should Adjuvant Chemotherapy Be Administered? What Is the Recommended Schedule? How Should Postoperative Chemotherapy and Radiation Be Combined?

Recommendations:Adjuvant chemotherapy constitutes a standard option of treatment for patients with high-grade, deep, and >5 cm localized STSs (I,A), especially if they are located in the extremities and trunk wall (II,A). Adjuvant therapy is not recommended for retroperitoneal and visceral sarcomas (II,B).However, close observation without chemotherapy could also be an option, especially in patients with a high risk of associated toxicity. The decision of whether to treat or not should be made in each individual case, after a discussion of potential benefits and risks with the patient (V,B).A regimen including anthracyclines and ifosfamide is recommended (II,A). For patients younger than 65, a regimen including high doses of both agents, such as that of Frustaci et al., is preferred (II,B). A total of three cycles of adjuvant chemotherapy seems to be enough (II,B). When adjuvant chemotherapy is administered, postoperative radiotherapy could be administered upon its completion (II,B).

Literature review and interpretation:

Adjuvant chemotherapy has been widely studied because the risk of a subsequent development of metastasis in a group of patients with high-risk STSs is close to 50%. However, several available randomized studies comparing chemotherapy to observation provide conflicting results. Variations in the chemotherapy schedules and the selection criteria used could partially explain this. A meta-analysis including a total of 18 phase III trials comparing adjuvant chemotherapy versus observation in resected localized STSs was conducted [84]. The results show that chemotherapy was associated with a risk reduction in overall recurrence of 10% and an improvement in overall survival of 6%. Furthermore, the benefits appeared to be greater when a combination of anthracycline and ifosfamide was administered. A previous meta-analysis, including only 14 trials, showed greater benefits in a group of patients with extremity sarcomas [85].

A recent trial, not included in the above-mentioned meta-analysis, involved patients with high- and intermediate-grade STSs at any site. The adjuvant chemotherapy consisted of doxorubicin and ifosfamide versus a placebo. No differences in survival between both arms were found [86]. Nevertheless, an update of the meta-analysis included in this study showed a difference in overall survival, favouring adjuvant chemotherapy with a hazard ratio of 0.86 (95% CI 0.79–0.97) [86]. Consequently, data from a meta-analysis indicate that adjuvant chemotherapy using anthracycline-based regimens provides a significant, although limited, improvement in relapse and survival in patients with high-risk STSs. Nevertheless, the toxicity that could be associated with these regimens, especially in frail or elder patients, suggests that the decision to administer adjuvant chemotherapy should be considered individually.

Although the classical recommendation for adjuvant chemotherapy consists of five cycles, the results of a recent randomized trial of neoadjuvant chemotherapy suggest that in a perioperative setting, a total of three cycles could be enough [80]. In the absence of further data, the EORTC 62,931 trial suggests that delaying the administration of radiotherapy until completion of adjuvant chemotherapy is not associated with worse local control [86], but it has the potential advantage of not compromising the optimal chemotherapy administration and reducing the risk of toxicity caused by the combination of anthracyclines and RT.

#### 3.2.7. Question: How Do We Treat an Isolated Local Relapse?

Recommendations:Limb-sparing surgery is the treatment of choice when achieving adequate margins, if possible; otherwise, amputation should be recommended (III,B).Perioperative radiotherapy must be considered if not previously administrated (II,A). In selected cases, re-irradiation can be considered (IV,B).

Literature review and interpretation:

A local relapse has been reported in approximately 7–15% of cases and might be present without evidence of systemic disease [87]. Upon diagnosis of a local relapse, an adequate workup including a thorax CT scan to rule out metastatic spread must be performed. For an exclusive local relapse, limb-sparing surgery is the treatment of choice when it is considered possible to achieve adequate margins and correct functional reconstruction of the extremity. Otherwise, amputation should be the preferred surgical approach [88]. After conservative surgery, radiotherapy must be administered if not previously performed after the first surgical intervention [73]. Re-irradiation is not a standard practice, although it could be considered for selected patients. These decisions should be made on an individual basis.

#### 3.2.8. Question: How Should Patients with STSs Be Followed After Their Primary Tumour Treatment?

Recommendations

A follow-up after primary treatment should be performed at 3- to 6-month intervals for the first 2 years and at 6-month intervals until the 5th year (III,B), tailoring according to tumour grade and other tumour- and patient-related factors (IV,B). The role of follow-ups after five years is unclear (V,C).For local control, history-taking and physical examinations should be a part of every follow-up visit (III,B). MRI scans are the imaging test for local control. Ultrasonography can be an alternative in superficial or palpable tumours (IV,C).Thoracic imaging should be conducted at every follow-up visit; a chest X-ray or a thoracic CT are the imaging tests of choice (III,B).Routine brain or extra-thoracic imaging is not recommended, except if there is a clinical suspicion of metastatic disease in these locations (V,C). Spine or whole-body MRI may be considered in myxoid liposarcomas. FDG-PET is not recommended as the 1st choice for the detection of local relapses and pulmonary metastases, although it can be useful in selected cases for the detection of extrapulmonary visceral spread (V,C).

Literature review and interpretation:

There is little consensus on the reasons for or in the frequency of follow-up examinations in patients with STSs. In addition, the overall duration of follow-ups and the most suitable imaging procedures have also not been conclusively established [89].

The risk of developing local or distant metastases is associated with numerous factors, such as the histological subtype, tumour grade, tumour size, surgical margins, (neo-adjuvant radiotherapy or chemotherapy), and other patient-related factors [90]. High-risk patients generally relapse within 2–3 years, whereas low-risk patients may relapse later, although this is still significantly less common than early recurrences. With all these caveats, a reasonable approach is that recommended above [89]. Some prospective data suggest that there is little difference between three- and six-month follow-up intervals [91]. On the other hand, more frequent visits might be indicated for certain high-grade sarcomas, such as paediatric sarcomas seen in the adult population [92]. Finally, although very late relapses can be seen, the role of a follow-up after five years remains unclear. Despite this, most studies and guidelines do advocate continuing a yearly follow-up from the 5th to the 10th year [89].

The follow-up modality for local and metastatic disease is also controversial. Some form of chest imaging is recommended in all follow-up visits. Both chest X-ray and thoracic CT scans are routinely used, and there is wide institutional and national variability in the use of one or the other of these modalities [93,94]. Although the early use of thoracic CT to scan for lung metastases is likely to pick up recurrences earlier, it has not been proven that this is beneficial or cost-effective compared with regular chest X-rays [91]. In any case, a chest CT scan should be obtained in the event of suspicious findings on the chest X-ray.

### 3.3. Section 3: Management of Metastatic Disease

#### 3.3.1. Question: When Should Surgery Be Considered for Metastatic Patients? Should It Be Combined with Systemic Treatment?

Recommendations:Metastasectomy should be considered, particularly in patients with pulmonary metastases, when the following criteria are met (IV,B): The primary tumour is controlled, without evidence of multiple metastatic disease at different organs.It should be considered in cases of metacronic pulmonary metastases, considering the interval between a surgical resection of the primary tumour and the development of metastases (>1 year) and the ability to achieve a complete resection.In rare histologies, without evidence of any benefit of systemic treatment, surgery should be considered the preferred option (IV,C).There is no evidence of the benefit of systemic therapy after a complete resection of metastasis; thus, it is not routinely recommended (IV,C).

Literature review and interpretation:

In STSs, the lung is the most frequent site of metastases, although some histologies (such as myxoid/round cell liposarcomas) can develop extrapulmonary metastases [95]. A majority of these cases are incurable, but in some selected cases, a surgical resection of metastatic disease can improve relapse-free survival or be potentially curative [96,97]. A surgical resection is usually recommended in cases of long intervals between surgeries and a low number of metastasis (oligometastatic disease). Retrospective studies have analysed several factors, but the most important is the ability to resect all lesions [98]. The benefit of chemotherapy after pulmonary metastasectomy is unproven and is based on retrospective studies [99]. As a consequence, it cannot be routinely recommended and could be considered in selected cases and chemosensitive histologies, no prior chemotherapy, or synchronic metastases with surgery of the primary tumour and also metastasectomy.

#### 3.3.2. Question: What Is the Role of Stereotactic Body Radiotherapy (SBRT) in Metastatic STSs?

Recommendations:In oligometastatic patients with both pulmonary and extrapulmonary metastases, SBRT should be considered, particularly for lesions located in sites that are unresectable or where surgery would entail significant morbidity (III,B).

Literature review and interpretation:

In oligometastatic patients, SBRT demonstrates excellent control rates for both pulmonary and extrapulmonary metastases. It achieves control rates of ≥80% in cases of oligometastatic disease confined to the lungs and may offer similar effectiveness to surgery in terms of overall survival, while maintaining a favourable toxicity profile [100]. Additionally, SBRT enables the delivery of effective treatment to anatomical sites that are not amenable to a surgical resection [101,102].

#### 3.3.3. Question: What Is the Role of Ablative Focal Treatments Like Cryotherapy and Radiofrequency in Advanced/Metastatic Disease?

Recommendations:In oligometastatic disease, imaging-guided percutaneous ablative focal treatments may be indicated in patients with small volume and bilateral pulmonary disease, in which a surgical resection is not an option; in cases of multiorgan involvement where radical treatment is planned for all sites, but there is a desire to avoid multiple major resections (IV,B); and in patients with locally advanced disease, who are unable to tolerate other treatments (IV,B).

Literature review and interpretation:

Percutaneous ablative focal treatments provide a good treatment option with excellent local control for selected patients with oligometastatic STSs. Several single-centre retrospective series of pulmonary metastases treated with percutaneous ablation [103] reported high local control rates. Complete ablation of the treated pulmonary lesions [104], a disease-free interval after ablation >12 months [105], and a tumour size of 1 cm or smaller [106] were associated with better prognosis. Patients with recurrent sarcomas who were not surgical candidates [107] and patients with oligometastatic disease who had been stable on chemotherapy and had residual disease (lung or retroperitoneal nodes) [108] could benefit from ablation therapy (radiofrequency or cryoablation).

#### 3.3.4. Question: What Is the Best First-Line Option for Metastatic Patients Who Are Not Candidates to Surgery?

Recommendations:Anthracycline monotherapy is the first-line standard treatment for most metastatic patients that are not candidates for local treatment (I,A).Combination therapy with anthracycline plus ifosfamide should be considered for patients who may benefit from tumour reduction for symptom palliation or improving resectability (I,B).Doxorubicin plus trabectedin should be considered for fit leiomyosarcoma patients (I,A).Other options could be considered for specific subtypes, such as high-dose ifosfamide for synovial sarcomas (III,A) or weekly paclitaxel for angiosarcomas (III,B).

Literature review and interpretation:

Anthracyclines (doxorubicin 75 mg/m^2^ or equivalent) are still the standard first-line treatment for most advanced STS patients, offering a response rate (RR) of 8–14% and progression-free survival (PFS) of 6.8–7.1 months in modern trials [109]. Although a combination of doxorubicin plus ifosfamide versus (vs) doxorubicin alone increased the RR and PFS in sensitive histological types, it also increased toxicity and did not improve overall survival (OS) [110]. Thus, this combination is reserved for patients who may benefit from tumour reduction for symptom palliation or improving resectability, with an ECOG of 0–1 and under 65 years, trying to prevent excessive toxicity.

Recently, a randomized phase III trial with the combination of 60 mg/m^2^ of doxorubicin followed by 1.1 mg/m^2^ of trabectedin vs. 75 mg/m^2^ of doxorubicin in patients with leiomyosarcomas reported a median PFS of 13.5 vs. 7.3 months, an RR of 38 vs. 13%, and a median OS of 30.5 vs. 24.1 months, respectively [111]. If available, it should be considered as a first-line option for these patients with a good performance status.

High-dose ifosfamide (>12 g/m^2^) is an alternative for synovial sarcomas, as this subtype is especially sensitive to ifosfamide, with an RR > 40% [112]. In a single-arm phase 2 trial and retrospective reports, weekly paclitaxel offered an RR of 20–40% and a PFS of 4–5.6 months for angiosarcoma patients [113,114]. Thus, it is a clear alternative for these patients in the first line, especially in scalp angiosarcomas. No other drugs added to doxorubicin (evofosfamide, palifosfamide, and olaratumab) or alternative combinations (gemcitabine plus docetaxel) have shown an advantage in overall survival in recently randomized trials.

#### 3.3.5. Question: What Are the Alternative First-Line Therapies for Patients Who Are Not Eligible for Anthracyclines?

Recommendations:Ifosfamide (III,A) and pegylated liposomal doxorubicin (IIB) are alternatives when anthracyclines are contraindicated because of prior treatment with anthracyclines in the neo/adjuvant setting or during the treatment of another cancer.Other options in this setting are approved drugs for a second line and beyond, like pazopanib, trabectedin, and DTIC–gemcitabine (see section on second lines) (IV).Metronomic oral cyclophosphamide plus prednisolone could be an alternative for patients >65 years (III,B).

Literature review and interpretation:

As previously stated, ifosfamide is an alternative for synovial sarcomas, as this subtype is especially sensitive to ifosfamide. Pegylated liposomal doxorubicin (50 mg/m^2^ of Caelyx) showed good performance when compared to 75 mg/m^2^ of doxorubicin in a phase II trial [115]. The RR was 10 vs. 9%, respectively, with significantly less neutropenia and alopecia but a higher incidence of palmar–plantar erythrodysesthesia [116].

A total of 100 mg of metronomic oral cyclophasmide twice daily plus 20 mg of prednisone daily has been tested in patients >65 years in a small non-randomized study [117]. The reported RR was 26%, and the median PFS was 6.8 months, and the toxicity profile was favourable with few discontinuations.

#### 3.3.6. Question: What Systemic Treatment Options Are Available for Advanced STS Patients in the Second and Subsequent Lines? In Which Cases and at What Point Should Each Option Be Administered?

Recommendations:For low-grade or poorly chemotherapy-sensitive sarcomas, especially in asymptomatic patients, active surveillance may be a good option (IV,C).Second and further lines in advanced disease have a palliative objective and should mainly be considered in fit and symptomatic patients (I,B).There are several treatments in this setting, and mostly, the decision is based on histology, toxicity profile, and patient preferences (IV,C): -Trabectedin should be considered for the treatment of patients diagnosed with all sarcoma subtypes after progression or who are ineligible for doxorubicin and ifosfamide [IIB]. There is more evidence of drug activity in leiomyosarcomas and liposarcomas (L-sarcomas) [I, A]. In myxoid liposarcomas, it achieves a high response rate and should be given as the first choice until progression (IV,A).-Pazopanib is indicated for the treatment of pre-treated patients diagnosed with non-adipocytic sarcomas (I,A).-Eribulin constitutes an alternative in the treatment of liposarcomas after progression to anthracycline (I,A).-Although both combinations of gemcitabine with dacarbazine or docetaxel are active in the treatment of these tumours, the former has a better tolerance profile and is, therefore, the best option, being especially useful in leiomyosarcomas [II,B].-Ifosfamide at doses higher than 9 g/m^2^ in patients who have not received it in the first line, or at high doses (>10 g/m2), subsequent to standard doses constitutes an option, especially in synovial sarcomas [III,B].For the majority of STSs, there is no evidence that a particular sequence is better than another, and most patients with a good performance status will probably benefit from being exposed to the largest number of available drugs (IV,C).

Literature review and interpretation:

Several effective treatment options in this situation have not been directly compared; therefore, treatment selection will be determined by histologies, patient preferences, and tolerance profiles. Despite this, some histological subtypes have shown greater sensitivity to certain drugs that may help us make a decision. This is the case of synovial sarcomas with high-dose ifosfamide, L-sarcomas with trabectedin, and leiomyosarcomas with gemcitabine combinations.

Trabectedin (1.5 mg/m^2^ in a 24 h infusion/21 days) has been broadly explored in several phase I–II trials. Although the RR was modest, a higher progression arrest rate was observed, especially in liposarcoma (notably myxoid liposarcoma, 88%), synovial sarcoma, and leiomyosarcoma, but also in other tumour types. A phase III trial, including pre-treated patients with liposarcomas or leiomyosarcomas, demonstrated better PFS with trabectedin over dacarbazine monotherapy [118]. Pazopanib (800 mg daily) improved PFS and disease stabilization versus a placebo in a randomized double-blind phase III study that enrolled patients with a non-adipocitic sarcoma progressing to chemotherapy [119]. Eribulin (1.4 mg/m^2^ days 1 and 8/21 days) has been approved for liposarcomas based on a phase III trial comparing eribulin vs. dacarbazine after progression to anthracycline. Although OS was significantly superior in the entire population, this was due to the effectiveness of the drug in the liposarcoma subgroup in which a 7-month gain in OS was achieved [120].

The superiority of the combination of gemcitabine (1800 mg/m^2^ at 10 mg/m^2^/min) with dacarbazine (500 mg/m^2^) every 14 days versus dacarbazine alone has been reported in a randomized phase II trial in terms of PFS and OS, with a favourable toxicity profile. Although activity was observed in different histological subtypes, the treatment was especially active in leiomyosarcomas [121]. Gemcitabine (900 mg/m^2^ at 10 mg/m^2^/min days 1 and 8) in combination with docetaxel (100 mg/m^2^ at day 8) every 21 days was more effective in terms of PFS, OS, and RR than gemcitabine monotherapy in a multicentric randomized phase II trial, especially in patients with leiomyosarcomas and undifferentiated pleomorphic sarcomas. This benefit was at the expense of higher toxicity [122]. A dose–response relationship has been described for ifosfamide, and, therefore, those patients who have received conventional doses in the first line may benefit from higher doses (≥12 g/m^2^ every 28 days) after progression. This strategy is especially useful in synovial sarcomas [123].

#### 3.3.7. Question: Is There Any Role for Immunotherapy in STSs?

Recommendations:Immunotherapy with anti-PD1/PDL1 is not standard in any sarcoma subtype but can be strongly recommended in selected subtypes (III,B).Alveolar soft part sarcomas (ASPSs) and malignant rhabdoid tumours represent the most sensitive sarcoma subtypes to anti-PD1/PDL1 compounds. The compassionate use of these compounds is strongly recommended [124,125] (III,A).Other sarcoma subtypes that have demonstrated signs of activity with anti-PD1/PDL1 in prospective studies are undifferentiated pleomorphic sarcomas, dedifferentiated chondrosarcomas, dedifferentiated chordomas, angiosarcomas, and epithelioid sarcomas. The option of clinical trials with anti-PD1/PDL1 is particularly recommended in these patients [126] (III,B).Adaptive immunomodulation with TCR-T cell therapy against sarcomas expressing neoantigens (NY-ESO1 and MAGEA4) as synovial sarcomas or myxoid liposarcomas is a promising approach, showing responses in up to 50% of cases. However, this is not a standard treatment, and inclusion in clinical trials is strongly recommended [127,128] (III,B).

Literature review and interpretation:

Several clinical trials have been published exploring the impact of anti-PD1/PDL1 in sarcomas. In monotherapy with pembrolizumab or nivolumab, the 6-month PFS rate and the overall response rate seem inferior to combinations of these agents with ipilimumab or antiangiogenics (axitinib and sunitinib). Within the most frequent sarcomas, an undifferentiated pleomorphic sarcoma is related to a higher probability of response, whereas some uncommon and ultrarare sarcomas mentioned above exhibit higher sensitivity to response and disease control.

### 3.4. Section 4: Therapeutic Considerations for Specific Subtypes of STSs

#### 3.4.1. Question: What Are the Diagnostic and Therapeutic Peculiarities of Retroperitoneal Sarcomas (RPSs)?

Recommendations:These patients should be managed by expert surgeons at referral centres with multidisciplinary units and oncology committees [III,A]. Chest–abdominal CT scan and extraperitoneal core needle biopsy are the procedures of choice for primary diagnosis [IV,A].An en bloc resection of the tumour, including adjacent organs, is the only curative treatment for RPSs [III,A].Postoperative radiotherapy after a complete gross resection should not be used in retroperitoneal sarcomas (RPSs) due to a lack of evidence of benefits [I,B]. Preoperative radiotherapy is not a standard treatment but could be considered in selected patients with low-grade liposarcomas in a multidisciplinary sarcoma tumour board [IV,D].Adjuvant and neoadjuvant chemotherapy should not be routinely employed in RPSs due to a lack of evidence of benefits [IV,D]. Neoadjuvant chemotherapy may be considered in the case of technically unresectable/borderline resectable RPSs and chemosensitive histologies [IV,B].Surgery of local recurrences should be considered, especially in cases with a disease-free interval of more than 6 months between an initial resection and recurrence [V,B].Treatment of advanced disease is similar to all other locations [II,B].

Literature review and interpretation:

Surgery is the only potentially curative treatment and should be performed by expert surgeons in high-volume sarcoma treatment centres with multidisciplinary committees. A complete en bloc gross resection, to achieve negative margins, offers the best chance for long-term survival for RPSs. However, adequate surgical margins, the principal prognostic factor, are rarely obtained. The surgical approach should be tailored to the histologic subtype [129].

The role of neoadjuvant RT was evaluated in a randomized phase III study (STRASS) in which it did not have an impact on local control or overall survival in the overall study population. However, an improvement in abdominal recurrence-free survival, the primary endpoint, was targeted in the LPS subgroup. Based on their results, the routine use of neoadjuvant RT is not recommended in patients with high-grade LPSs and LMSs but may be considered in low-grade LPSs [130].

Surgery of local recurrences should be considered, especially in cases with a disease-free interval after a previous resection that is greater than 6 months or a growth rate of less than 1 cm per month [131].

#### 3.4.2. Question: What Is the Recommended Treatment for Desmoid Tumours?

Recommendations:Active surveillance should be the first approach after diagnosis in the majority of patients (IIIA). Active treatments should only be considered in case of persistent progression or if the disease is located in a critical location (IIIA).In case of progression (radiological or clinical), surgery can be proposed if the tumour is located in the abdominal wall. For all other locations, medical treatment is usually preferred (IIIA).Medical treatment options, along with their evidence levels, are as follows: nirogacestat (I,A), sorafenib (I,A), pazopanib (II,B), imatinib (III,A), chemotherapy (III,A), and anti-hormonal therapies or NSAIDs (IV,B)Other treatments include cryoablation, radiotherapy, and surgery (IIIB), which should be evaluated in a case-by-case scenario.

Literature review and interpretation:

Desmoid tumours often have an unpredictable course. Spontaneous regressions can occur in up to 20–30% of cases, and, in retrospective series, a progression-free survival of 50% at 5 years in patients with no active treatment has been reported [132]. Based on these facts, upfront surgery has been substituted by active surveillance as the standard of care for the majority of patients.

There is no data to guide the best initial treatment when a tumour progresses or is symptomatic. Tumour location is a risk factor for a local relapse after surgery [133]. The abdominal wall is the most favourable site for surgery as the primary treatment. For other locations, surgery is not the first choice but may be considered if expected morbidity is low. Several agents of medical therapy are available, and treatment choices should be made based on the location of the tumour, growth speed, the need of a response, expected toxicity, and the patient’s age and preferences. In a recent randomized trial, nirogacestat showed high activity in terms of response (41 vs. 8%) at 2 years in PFS (76 vs. 44%) and clinical improvement in a phase III trial compared to a placebo [134]. Sorafenib was also evaluated in a phase III placebo-controlled trial, yielding a 33% response rate [135]. Pazopanib was evaluated in a phase II randomized trial against chemotherapy, showing a PFS rate at 6 months of 83.7% compared to 45% for chemotherapy [136]. Chemotherapy options include low-dose methotrexate plus vinblastine [137] and liposomal doxorubicin, which is usually reserved for patients in need of a rapid response [138].

#### 3.4.3. Question: Which Peculiarities Does the Treatment of Gynaecological Sarcomas Have?

Recommendations:Surgery: en bloc total hysterectomy and bilateral oophorectomy. Lymphadenectomy is not routinely indicated. Morcellation is not indicated (II,A). Bilateral oophorectomy could be omitted in stage I low-grade endometrial stromal sarcomas (III,B).Adjuvant chemotherapy is not the standard of care in stage I (II,D). It should be considered in stages II and III in a case-by-case multidisciplinary discussion (II,C). Adjuvant radiotherapy is generally not recommended (II,C).Adjuvant chemotherapy and first-line advanced disease: anthracycline-based chemotherapy (I,A).Stage IV low-grade endometrial stromal sarcomas: consider hormonal treatment as a first-line treatment (II,A).

Literature review and interpretation:

Gynaecological sarcomas include different subtypes: leiomyosarcomas (LMSs) (63%), endometrial stromal sarcomas (ESSs) (21%), and high-grade [139] uterine sarcomas (HG-USs). Carcinosarcomas behave as epithelial carcinomas and are not covered in these guidelines. Adjuvant chemotherapy is not the standard of care in stage I uterine sarcomas (USs) but may be considered individually in stages II and III. Four courses of adjuvant gemcitabine/docetaxel followed by four courses of doxorubicin versus no adjuvant treatment have been tested in a phase III study with a premature closure due to lack of accrual [140]. Thus, the adjuvant regimen in USs could be parallel to other soft tissue sarcomas (STSs) where anthracycline-based chemotherapy is standard. For metastatic USs, the treatment mimics that for STSs. Gemcitabine/docetaxel has not shown a survival benefit compared with doxorubicin [141]. Adjuvant radiotherapy is generally not recommended as it has failed to demonstrate a survival and local control benefit [142]. Hormonal therapy (megestrol acetate/aromatase inhibitors/gonadotropin-releasing hormone (GnRH)) is preferred as a first-line treatment in oestrogen receptor-positive stage IV LG-ESSs [143].

#### 3.4.4. Question: What Sarcoma Pathological/Molecular Histotypes Require a Specific Approach?

Recommendations:

Among all sarcoma classification lists, there are some uncommon subtypes whose treatment differs from the most prevalent ones. Here, relevant aspects that must be addressed differently from the general guidelines are highlighted. Table 1 shows the diagnostic and therapeutic recommendations for the advanced disease that should be taken into account when these tumours are approached [144].

## 4. Conclusions

These guidelines underscore the critical importance of a multidisciplinary, evidence-based approach to the diagnosis and management of STSs. By integrating the latest advancements in molecular biology, imaging technologies, and treatment modalities, the guidelines aim to offer a comprehensive framework for clinicians, enhancing diagnostic accuracy and optimizing therapeutic strategies. The collaborative efforts of specialists across various fields are essential to ensure individualized, high-quality care for STS patients, ultimately improving both survival outcomes and quality of life. Furthermore, it is recommended that STS patients be treated at expert reference centres with specialized experience in sarcoma management, as this can contribute to improve treatment outcomes. Finally, we would like to highlight the importance of strengthening clinical and translational research on this complex disease to ensure further progress in the near future.

## Figures and Tables

**Table 1 cancers-17-03158-t001:** Uncommon sarcoma subtypes with a specific treatment approach.

Histotype	Recommended Molecular Diagnostic Tests	Therapeutic Recommendations	
Angiosarcomas		-1st line CT: anthracyclines or taxanes. -2nd: gemcitabine or pazopanib.	II/B III/B
Alveolar soft part sarcomas	*ASPSCR1-TFE3* gene fusion	-1st line: anti-angiogenics or/and immunotherapy	II/B
Dermatofibrosarcoma protuberans	*COL1A1-PDGFB1* gene fusion	-1st line: imatinib. -2nd line: pazopanib.	II/A II/B
Clear cell sarcomas	*EWSR1-ATF1* gene fusion	-1st line: anti-angiogenics	II/B
Solitary fibrous tumours	*NAB2/STAT6* gene fusion	-1st line: TKIs, pazopanib, or axitinib. -2nd line: conventional CT. -3rd line: Temozolamide–bevacizumab.	II/A III/B IV/B
Inflammatory myofibroblastic tumours	ALK; ROS1 and NTRK gene rearrangements	-1st line, ALK positive: crizotinib. -1st line, ALK negative, and 2nd line, ALK positive: anthracycline/Vinca alcaloids +methotrexate	II/A IV/B
Epithelioid haemangioendotheliomas	*WWTR1-CAMTA1* gene fusion [6]	-1st line: anti-angiogenic	IV/B
Kaposi sarcomas	HHV8 related	AIDS-related KS Treatment: -No visceral involvement: antiretroviral therapy alone. -Immune reconstitution inflammatory syndrome: CT. CT advanced disease (general): -1st line: pegylated doxorubicin. -2nd line: paclitaxel.	II/A IV/A I/B II/A
Epithelioid sarcomas	*SMARCB1 (INI1)* deficient	-1st line CT: tazemetostat (if available). -2nd: anthracycline-based CT.	III/A IV/A
Tenosynovial giant cell tumours	*COL6A3-CSF1* gene fusion	-Diffuse type: -CSF1R inhibitors (if available). -imatinib.	II/A IV/A
Extraskeletal myxoid chondrosarcomas	*NR4A3* rearragements	-1st line: pazopanib (II, A) -2nd line: anthracycline-based CT	II/A
PEComas	TSC1/TSC2 gene alterations [6]	-1st line: mTOR inhibitors. -2nd line: anthracycline-based CT.	II/A IV/B
NTRK-rearranged sarcomas	NTRK rearrangements	-1st line: NTRK inhibitors	III/A

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
