# Peer review of "Diagnosis and Therapy of Soft Tissue Sarcomas: Spanish Group for Research in Sarcomas (GEIS) Guidelines"

_cancers, 2025, doi:10.3390/cancers17193158_

Round 1

Reviewer 1 Report

Comments and Suggestions for Authors The paper aims to provide guidelines for the treatment of soft tissue sarcomas. Although this is not an original topic, well-structured reviews are highly relevant, especially in the heterogeneous field of sarcomas. This paper provides a good overview of the current data available in the interdisciplinary field of sarcoma treatment. In addition, its structure, with specific, clinically relevant questions and well-structured answers backed up by adequate sources, makes it very easy to read. The methodology is appropriate for the type of publication, and the answers have already been peer-reviewed by the experts involved in the paper's creation process. The conclusions and answers to the individual questions are therefore well thought out and can be considered very appropriate. The sources cited are very comprehensive and well chosen. The attached table on possible systemic therapies/immunotherapies also offers significant added value, is very clear, and is an asset to everyday clinical practice. I consider the paper to be a very good compilation of the current data on sarcomas, which is definitely worthy of publication in this form.

Author Response

Se sincerely thanks the reviewer for their thorough evaluation and highly encouraging comments regarding our manuscript.

We are delighted that the reviewer recognizes the clinical relevance and practical utility of our work, particularly in the context of sarcoma´s heterogeneity and the need for clear, multidisciplinary guidance. We greatly appreciate the positive assessment of the manuscript´s structure, as well as the methodology, source selection, and the value of the therapeutic table included.

Reviewer 2 Report

Comments and Suggestions for Authors

Thank you for the opportunity to review this manuscript

First of all I would like to acknowledge the authors for the fine work herein presented. Within, the authors display some guidelines for STS diagnosis and management based on the available evidence, also considering the most recent findings in molecular biology and imaging.

This manuscript offers a sum of comprehensive recommendations for STS approach and management, being very well written and easy to follow. There are virtual no errors detectable within the manuscript, presenting state of the art objective guidelines for those who deal directly and indirectly with soft tissue sarcomas.

In the end, the review presented is well designed and presented, with the recommendations always being supported with the most relevant evidence available in literature.

This work is robust and despite not presenting novelty regarding soft tissue sarcoma approach and management, has an enormous value  in summarizing fundamental information to facilitate the management of such difficult clinical entities.

Congrats to all

Author Response

Thank you very much for your thoughtful and encouraging review of our manuscript, “Diagnosis and Therapy of Soft Tissue Sarcoma: Spanish Group for Research on Sarcomas (GEIS) 2025 Guidelines.”

We sincerely appreciate your recognition of the manuscript’s clarity, structure, and clinical utility. Your acknowledgement of the effort to synthesise up-to-date, evidence-based recommendations — integrating advances in molecular biology, imaging, and multidisciplinary care — is significant to our team.

Reviewer 3 Report

Comments and Suggestions for Authors

With this work, the authors are providing comprehensive clinical guidelines for soft tissue sarcomas. They have addressed the  most important aspects in the diagnostic and treatment of this group of malignancies. The authors made an impressive job with preparing clear, concise and well-documented recommendations for the diagnostic and treatment of STS. The list of references is also impressive. My only comments concern the introduction part which is kind of scarce in information. For instance, a classification of the STS in the instructive part would be useful for the reader. Overall, the article is excellent and my strong recommendation is  to be accepted for publication.

Author Response

We thank the reviewer for their positive assessment and valuable suggestions. As recommended, we have enriched the introduction by incorporating a brief classification of soft tissue sarcomas (STS), according to the World Health Organization. This addition provides context for the heterogeneity of these tumours and helps readers better understand the diagnostic and therapeutic challenges discussed throughout the guidelines.

We have carefully considered all comments and believe the manuscript has been significantly strengthened. Thank you again for your consideration.

Reviewer 4 Report

Comments and Suggestions for Authors

This paper offers a thorough and strong collection of evidence-based guidelines for diagnosing and treating soft tissue sarcomas.The multidisciplinary, consensus-driven method is a major strength since it makes sure that the suggestions are therapeutically relevant and trustworthy.The document is well-organized, current, and will definitely be a useful tool. 
 The quality is great overall; however, a few small changes may make it even better.First, putting a "Key Points" section at the beginning would help readers rapidly understand the most important clinical ideas.The abstract is well-written; however, it may be better if it summarized one or two of the most important recommendations instead of just saying what the study is about.

Author Response

We thank the reviewer for their positive assessment and highly valuable suggestions. We have implemented both changes as recommended:

  1. A 'Key Points' section has been added at the beginning of the manuscript to provide a concise overview of the core clinical recommendations.

  2. The abstract has been revised to include specific guidance, now highlighting the critical importance of MRI as the preferred initial imaging modality and mandatory referral to a specialized sarcoma center with a multidisciplinary team.

We believe these changes significantly enhance the manuscript's clarity and immediate impact for the reader